# Hepatic Proteomic Analysis Reveals That Enhanced Carboxylic Acid Metabolism and Oxidoreduction Promote Muscle and Fat Deposition in Muscovy Duck

**DOI:** 10.3390/ani11082180

**Published:** 2021-07-23

**Authors:** Wanli Yang, Xingyong Chen, Congcong Wei, Yutong Zhao, Zhengquan Liu, Zhaoyu Geng

**Affiliations:** 1College of Animal Science and Technology, Anhui Agricultural University, Hefei 230036, China; 18435131403@163.com (W.Y.); 17681322537@163.com (C.W.); z18255494174@163.com (Y.Z.); zliu13155@gmail.com (Z.L.); gzy@ahau.edu.cn (Z.G.); 2Anhui Province Key Laboratory of Local Livestock and Poultry Genetic Resource Conservation and Bio-Breeding, Anhui Agricultural University, Hefei 230036, China

**Keywords:** Muscovy duck, liver protein, fat deposition, muscle fiber, two-dimensional electrophoresis, oxidoreduction

## Abstract

**Simple Summary:**

Liver plays an important role in lipid synthesis and muscle growth in poultry. The current study measured the growth traits and the proteome of Muscovy duck liver at 14, 28, 42, and 56 days, aiming at exploring the key regulatory proteins for intramuscular fat deposition and muscle growth. The results showed that Muscovy duck grew most rapidly at 28 vs. 42 days of age, subcutaneous and abdominal fat were deposited rapidly, but intramuscular fat content decreased. At the same time, the abundance of liver proteins regarding the tricarboxylic acid cycle and oxidoreduction increased significantly. This study provides a profile of the fat deposition and liver proteome for Muscovy duck.

**Abstract:**

Liver is responsible for 90% of lipid synthesis in poultry; thus, it plays an important role in the growth of Muscovy ducks, which have a high fat deposition ability in a time-dependent manner. Therefore, male Muscovy ducks at 14, 28, 42, and 56 days were selected for body weight (BW), carcass weight (CW), subcutaneous fat thickness (SFT), abdominal fat weight (AFW), intramuscular fat content (IMF), and breast muscle fiber (BMF) diameter and density determination. Two-dimensional electrophoresis (2-DE) combining liquid chromatography linked to tandem mass spectrometry (LC-MS/MS) was used to analyze proteomic changes in liver at each stage. The BW, CW, AFW, SFT, and BMF diameter and density were significantly increased, while IMF content was significantly decreased at 28 to 42 days of age (*p* < 0.05). There were 57 differentially abundant protein (DEP) spots representing 40 proteins identified among the ages, in which 17, 41 and 4 spots were differentially abundant at 14 vs. 28, 28 vs. 42, and 42 vs. 56, respectively. Gene Ontology enrichment analysis found that DEPs were mostly enriched in the oxidation-reduction process, carboxylic acid metabolism, etc. Protein–protein interaction showed that catalase (CAT), triosephosphate isomerase (TPI), and protein disulfide-isomerase (PDI) were the key proteins responsible for the growth of Muscovy duck. In conclusion, 28 to 42 days of age is the crucial period for Muscovy ducks, and the ability of metabolism and antioxidants were significantly enhanced in liver.

## 1. Introduction

In the breeding of meat poultry, people are more enthusiastic about the fast-growing and high intramuscular fat, but less about abdominal and subcutaneous fat. Higher intramuscular fat (IMF) is always associated with better taste [1], and less abdominal and subcutaneous fat can reduce feed consumption, shorten the feeding cycle, and improve feed conversion efficiency [2]. Being different from fast-growing meat duck, Muscovy duck shows a slower growth rate and fat deposition before 6 weeks of age, while a higher body weight after 10 weeks of age [1,3]. The intramuscular fat content of Muscovy duck is lower than that of Pekin duck, independent of overfeeding or not [4]. The slower growth rate leads to a late marketing age of Muscovy duck [5]. Therefore, how to improve the early growth rate is crucial for Muscovy duck breeding.

As for body fat, about 90% of lipids are synthesized by liver in poultry, and then, this enters the blood through apolipoproteins and is deposited in surrounding tissues. In overfed geese as well, the amount of synthetic fat is much more than the amount of fat transported, causing a large accumulation of fat in liver to form a fatty liver [6]. Thus, liver metabolism at different stages would reflect the fat metabolism. It has been suggested that Pekin duck has a stronger ability to transport fat, while Muscovy duck prefers to retain fat in liver [7]. Muscovy duck exhibits a similar hepatic composition as Pekin duck before overfeeding, and then, a higher degree of hepatic steatosis and a lower increase in adiposity occurs after overfeeding [8]. According to Hérault et al. [9], there are significant differences in the fatty acids and triacylglycerol synthesis and glycolysis between Pekin duck and Muscovy duck at the transcriptional level of liver. The intramuscular fat content was determined by the relative amount of muscle fiber and fat deposition [10]. 

Two-dimensional electrophoresis (2-DE) is a traditional proteomic technology. For species with no available protein library, 2-DE is more useful as the differential protein can be visually identified. Francois et al. 2014 used 2-DE combined with LC-MS/MS and found that heavier livers show higher expression of stress response proteins in mule ducks [11]. Zeng et al. 2013 [12] found 61 different spots and 54 proteins in liver between Pekin duck and Muscovy duck related to heat stress alleviation by using 2-DE combined with MALDI-TOF/TOF MS. Zheng et al. 2012 [13] identified 59 differential liver proteins of lean Pekin ducks at different ages by using 2-DE combined with MS, in which 37 upregulated proteins in adult ducks were involved in metabolism, defense, and antioxidation. Furthermore, 76 differentially abundant proteins responsible for fat metabolism were identified in liver between high- and low-fat Pekin ducks [14]. The number of fat cells is determined after birth and then expanded with lipid filling, which is synthesized by liver, thus increasing the fat content in tissues. Therefore, the proteomic changes of liver tissue in Muscovy duck at early growth stages are analyzed by 2-DE combined with liquid chromatography linked to tandem mass spectrometry (LC-MS/MS) to explore the key proteins that affect the early fat metabolism. This study aims to explore the characteristics of fat and muscle growth in Muscovy ducks and the metabolism changes in liver.

## 2. Materials and Methods

### 2.1. Ethics Statement

The animal experiment was reviewed and approved by the Institutional Animal Care and Use Committee of Anhui Agricultural University No. SYDW-P20190600601). The experiments were performed under the Regulations for the Administration of Affairs Concerning Experimental Animals and the Standards for the Administration of Experimental Practices, and the ARRIVE Guidelines 2.0 were followed.

### 2.2. Experimental Animals and Sample Collection

The male Muscovy ducks were reared and provided by Anqing Yongqiang Agricultural Technology Co., Ltd. Feed (Anqing, China) and water were provided ad libitum, and the nutrition level was under the National Research Council (NRC) standard [15]. Male ducks were raised in net-rearing system, 6 per square meter. The microclimate parameters were: 1–3 days old, temperature 30–32 °C, humidity 60–70%; 4–7 days old, temperature 28 °C, humidity 60–70%; 8–14 days old, temperature 25 °C, humidity 50–60%; 15–56 days old, temperature 20 ± 2 °C, humidity 50–60%. During the whole process, 24 h of light (10 Lux) were used. In the feed, crude protein (CP) accounted for 16%. The metabolic energy (ME) was 12.34 MJ/kg, and the energy to protein ratio was 12.97 g/MJ. At 14, 28, 42, and 56 days of age, 100 ducks were randomly selected from the whole group at each period to be weighed, and the average weight was calculated. In each period, 8 ducks with close to the average weight were selected. Body weight was recorded at each sampling time point after 12 h of fasting, and then, the ducks were slaughtered. Breast muscle on the left side was collected and stored in tissue fixative (4% paraformaldehyde). Breast muscle on the right side was stored at −20 °C for intramuscular fat content determination. The left lobe of liver was stored in liquid nitrogen and then transferred to −80 °C for protein extraction.

### 2.3. The Measurement of Growth Trait

Body weight (BW), carcass weight (CW), and abdominal fat weight (AFW) were registered by an electronic scale (ACS-300, HaoZhan, Guangdong, China). Subcutaneous fat thickness (SFT) near the pectoral muscle was measured by a vernier caliper. The IMF was measured by the Soxhlet extractor method (SOX406, HaiNeng, Shandong, China) [16]. After being fixed in the 4% paraformaldehyde for 24 h, the sample was washed with running water overnight. We dehydrated the tissue with 70–100% ethanol, then placed it in a xylene-ethanol (1:1) solution to become transparent. We soaked the transparent tissue in paraffin for 1–2 h to make paraffin blocks. We cut the paraffin blocks into 4μm-thick slices and put them on microslides. The hematoxylin-eosin (HE) sectioning of breast muscle was performed according to Xu et al. [17].

The total number of fibers was calculated (N) within three equal visions (S_1_) by the Olympus IX73 fluorescent inverted microscope (Olympus IX73, Tokyo, Japan). The area of each fiber (S_2_) was accurately measured, and the breast muscle fiber (BMF) diameter and density (ρ) of the muscle fiber were calculated according to the formula:d = 2 √ (S_2_/π)
ρ = N/S_1_

### 2.4. Liver Protein Extraction

The total protein was extracted by using the Total Protein Extraction Kit (MAO171, Meilun Biotech, Dalian, China) according to the protocol. Protein purification and salt removal were performed by using the acetone- trichloroacetic acid (TCA) method. In brief, 2 mL cold 10% TCA-acetone was added into 200 μL extracted protein in a centrifuge tube for precipitation for 12 h under −20 °C. After precipitation, the protein was washed with 10% TCA-acetone three times by centrifugation (10,000 rpm, 4 °C, 10 min) and then placed in a fume hood to completely evaporate the acetone. The purified protein was then completely dissolved in Buffer I (8 M urea, 4% CHAPS, 0.001% bromophenol blue). After quantification by using the BCA Protein Quantification Kit (20201ES76, Yeasen, Shanghai, China), the supernatant was stored at −80 °C for further utilization.

### 2.5. Two-Dimensional Electrophoresis

Eight samples from each time point were used for liver proteomic analysis by the 2-DE method, and each sample was analyzed three times by using the PROTEAN i12 IEF cell (Bio-Rad, California, CA, USA). The protein sample of 100 μg was added in Loading Buffer II (Loading Buffer I with 65 Mm Dithiothreitol (DTT), 0.2% Bio-Lyte) to a total volume of 125 μL, then loaded on an IPG strip (7 cm, pH 3–10, Bio-Rad, California, CA, USA) with 2 mL of mineral oil (Bio-Rad, California, CA, USA) coverage. The IPG was hydrated at 18 °C for 14 h for isoelectric focusing with the following procedures: 250 V for 0.5 h, 500 V for 0.5 h, 1000 V for 0.5 h, 12,000 V for 1 h, 3000 V for 1 h, 4000 V for 2 h, 4000 V for 20,000 Vh. After isoelectric focusing, the IPG strip was then equilibrated for 15 min in Equilibration Buffer I (6 M urea, 2% SDS, 0.375 M Tris-HCl, pH 8.8, 20% glycerol, and 0.1 M DTT) and for 15 min in Equilibration Buffer II (6 M urea, 2% Sodium dodecyl sulfate (SDS), 0.375 M Tris-HCl, pH 8.8, 20% glycerol and 250 mM iodoacetamide). Two-dimensional electrophoresis was performed by using 12% SDS polyacrylamide gel [18]. The gel was then stained for 30 min with Coomassie brilliant blue (P0017F, Beyotime, Shanghai, China) and then washed with double-distilled water three times.

### 2.6. Gel Analysis and Protein Spots Identification by LC-MS/MS

All gels were scanned and calibrated using a GS-900 Calibrated Densitometer (Bio-Rad, California, CA, USA). Protein spots were analyzed by PDQuest software (Version 8.0.1, Bio-Rad, California, CA, USA). The differentially abundant protein spots were cut out for enzymatic hydrolysis. The digested protein samples were redissolved in Nano-LC Mobile Phase A (0.1% formic acid) for identification by LC-MS/MS. The Easy nLC 1200 (ThermoFisher, Massachusetts, MA, USA) consisted of a C18 enrichment column (nanoViper, 3 μm, 100 Å), and a C18 analytical column (Acclaim PepMap RSLC, 75 μm × 25 cm, 2 μm, 100 Å) was used for the LC system. Mobile Phase B (80% acetonitrile, 0.1% formic acid) was set with a gradient increase from 5% to 38% within 10 min. The ThermoFisher Q Exactive combined with the Nano Flex Pump (ThermoFisher, Massachusetts, MA, USA) was used for the MS system, positive ion mode, Vcap of 1900 V, and drying gas temperature of 275 °C.

The original mass spectrometry data were analyzed by Mascot software 2.2 according to the duck protein database from UniProt (https://www.UniProt.org/, accessed on 9 September 2020) with the search parameters of trypsin digestion, cysteine iodoacetamide alkylation (+57 Da) as fixed modification, methionine oxidation (+16 Da), and asparagine and glutamine deamination (+0.98 Da) as variable modifications. The mass tolerance of the primary mass spectrum was ±20 ppm, and the second mass spectrum was ±0.05 Da. Protein identification was considered believable when the probability was greater than 95% and there were at least two identified peptides having maximal peptide coverage.

### 2.7. Bioinformatic Analysis and Statistical Analysis

One-way ANOVA using the software SPSS 22.0 (SPSS, Chicago, IL, USA) was used for BW, CW, AFW, STF, IMF, and different protein spots analysis. Gene Ontology (GO) analyses were performed by using the online database resource of DAVID 6.8 from https://david.ncifcrf.gov/home.jsp, accessed on 21 October 2020, and the GO terms of differential protein were clustered into biological processes, cell components, and molecular functions. For GO analysis, the protein spots at 14 days of age were set as the control when compared with 28 days of age (14 vs. 28), and proteins at 28 days of age were set as the control when compared with 42 days of age (28 vs. 42), which was also set as the control when compared with 56 days of age (42 vs. 56). The online website Heatmapper (http://www.heatmapper.ca/, accessed on 19 October 2020) was used to make the heatmap of the differentially abundant protein, and the data were normalized by using the z-value method. According to the duck protein database (https://www.UniProt.org/, accessed on 5 November 2020), the protein interaction network (PPI) was analyzed using the online database resource search tool (STRING 9.1, https://string-db.org/, accessed on 5 November 2020).

## 3. Results

### 3.1. Growth and Fat Deposition of Muscovy Duck at Different Ages

Within the experimental measuring stage, Muscovy duck showed continuous growth of body weight and carcass weight (*p* < 0.05, Figure 1A). The breast muscle fiber began to significantly grow at 28 vs. 42 (Figure 1), showing that the diameter of the BMF increased and the density of the BMF decreased quickly (*p* < 0.05, Figure 1A). As compared with the growth rate among different stages, the BW, CW, and BMF diameter all showed a high growth rate at 28 vs. 42 stage (Figure 1B).

The IMF content was not measured at 14 day of age because of the low meat production of Muscovy duck. The SFT and AFW increased from 1.62 mm and 3.31 g at 28 day of age to 2.50 mm and 28.23 g at 42 day of age (*p* < 0.05), respectively. However, the IMF decreased from 2.89% at 28 d of age to 1.15% at 42 day of age. From 42 to 56 day of age, for the significant increase of the AFW, no difference was observed in the IMF and SFT (Table 1). 

### 3.2. Differentially Abundant Proteins in Livers at Different Stages

The protein abundance showed a great increase from 28 to 42 day of age (Figure 2a) in liver of Muscovy duck. In total, 57 differentially abundant protein spots were identified among the ages (Figure 2b). At 14 vs. 28, there were 17 different spots, in which 12 spots (1, 5, 12, 28, 29, 30, 31, 32, 33, 35, 36, 38) were upregulated, 4 spots (34, 37, 39, 40) were downregulated, and 1 spot (57) newly appeared. At 28 vs. 42 stage, there were 41 different spots identified, in which 18 spots (1, 6, 7, 8, 9, 10, 11, 12, 13, 14, 15, 17, 18, 19, 20, 24, 26, 36) were upregulated, 8 spots (5, 16, 21, 22, 23, 25, 27, 56) were downregulated, and 15 spots (41, 42, 43, 44, 45, 46, 47, 48, 49, 50, 51, 52, 53, 54, 55) newly appeared. At the 42 vs. 56 stage, only 4 different spots were identified, in which 3 spots (2, 3, 4) were upregulated and 1 spot (1) was downregulated. Among different ages, Spots 1, 5, 12, and 36 repeatedly appeared.

### 3.3. LC-MS/MS Identification of Differentially Abundant Protein Spots

The 57 differentially abundant protein spots were identified by LC-MS/MS, and peptide matching was performed with reference to the *Anas platyrhynchos* protein library database. The 57 protein spots were annotated to 40 kinds of proteins (Table 2). There were 31 spots (1, 6, 7, 9, 10, 11, 12, 13, 14, 15, 16, 17, 19, 20, 25, 33, 34, 36, 42, 45, 46, 47, 48, 49, 50, 51, 52, 53, 54, 55, 56) representing 20 kinds of enzymes and the catalytic functions involved in the tricarboxylic acid cycle, sugar metabolism, cholesterol metabolism, folate metabolism, and antioxidant enzymatic hydrolysis. There were 8 spots (5, 8, 21, 28, 31, 35, 40) that represented 4 proteins, and their functions were not clear. The other 18 spots (2, 3, 4, 18, 22, 23, 24, 26, 27, 29, 30, 32, 37, 38, 39, 41, 43, 44, 57) represented 16 kinds of proteins as various structural and functional proteins during the development of Muscovy duck. Spots 6 and 7, which both were isocitrate dehydrogenase (IDH), had the same molecular weight, but different isoelectric points. Spots 10, 13, and 51 were dihydrolipoyl dehydrogenase (DLD). Their actual molecular weight and isoelectric point deviated from the theoretical value. Spots 34 and 46 and Spots 15 and 53, which were pyruvate dehydrogenase (PDH) and malate dehydrogenase (MDH), respectively, had the same molecular weight, but different isoelectric points. 

### 3.4. GO Enrichment for Differentially Abundant Proteins

The 17 differentially abundant protein spots were identified as 13 proteins at the 14 vs. 28 day stage (Table 3). They were significantly enriched in 43 biological processes, 29 cellular components, and 17 molecular functions (Figure 3). In biological processes, Peroxiredoxin 3 (PRDX3), Peroxiredoxin 1 (PRDX1), and TPI1 are enriched in the cellular response to oxidative stress (GO: 0034614). ACTG1 and Tropomyosin 4 (TPM4) are enriched in sarcomere organization (GO: 0045214) and myofibril assembly (GO: 0030239). In cellular components, proteins are mostly enriched in the myelin sheath (36%) and intracellular organelle (18%). In molecular functions, proteins are mostly enriched in protein binding (27%), oxidoreductase activity (18%), and thioredoxin peroxidase activity (18%, Appendix A). 

The 41 differentially abundant protein spots were identified as 39 proteins at the 28 vs. 42 day stage (Table 3). They were significantly enriched in 148 biological processes, 31 cellular components, and 64 molecular functions (Figure 4). In biological processes, 12 proteins are enriched in the oxidation-reduction process (GO: 0055114), 10 proteins are enriched in the carboxylic acid metabolic process (GO: 0019752), 10 proteins are enriched in oxoacid metabolic process (GO: 0043436), and 17 proteins are enriched in the single-organism catabolic process (GO: 0044712). In cellular components, proteins are mostly enriched in membrane-bounded vesicle (68%) and organelle (18%). In molecular functions, proteins are mostly enriched in oxidoreductase activity (43%) and catalytic activity (14%, Appendix A).

There were 4 differentially abundant proteins identified at the 42 vs. 56 day stage, Catalase (CAT), ATP synthase subunit beta (ATP5B), uncharacterized protein (ACTG1), and Zinc finger and BTB domain containing 22 (ZBTB22). In biological processes, CAT and ATP5B were significantly enriched in osteoblast differentiation (GO: 0001649) and ossification (GO: 0001503). In cellular components, CAT, ACTG1, and ATP5B were significantly enriched in extracellular membrane-bound organelle (GO: 0065010), extracellular exosome (GO: 0070062), extracellular vesicle (GO: 1903561), extracellular organelle (GO: 0043230), focal adhesion (GO: 0005925), cell-substrate adherens junction (GO: 0005924), cell-substrate junction (GO: 0030055), adherens junction (GO: 0005912), anchoring junction (GO: 0070161), and myelin sheath (GO: 0043209). There was no protein enriched in molecular function (Figure 5, Appendix A).

### 3.5. Protein–Protein Interaction Analysis of All Identified Differentially Abundant Proteins 

The protein network could provide insights into the biological process involving several proteins. Using the online tool STRING9.1, proteins at the node of the different biological interaction network (Figure 6) were considered as important regulators for growth and body fat synthesis in Muscovy duck. CAT, triosephosphate isomerase (TPI1), and protein disulfide-isomerase (PDIA3) had the highest degree of interaction with other proteins. These three key nodes clustered the proteins into three groups, in which twelve proteins were found to be related to the sugar metabolism pathway, nine to the redox reaction, and one to the chaperone. 

## 4. Discussion

### 4.1. Growth Performance of Muscovy Duck

As reported by Baeza et al. 2002 [19], French Muscovy duck showed a maximum daily gain at 28–49 days of age. In this study, Muscovy duck exhibited fast growth during 28 to 42 days of age. The deposition of lipids in abdominal and subcutaneous tissue increased before marketing age in poultry, but the duration was different. Subcutaneous fat deposition stopped at the earlier age of 42 days, while abdominal fat continued to accumulate with age. This trend has also been observed in other breeds. Murawska et al. (2012) [20] indicated that the subcutaneous fat of Pekin ducks remained stable at 28 days of age. Sonaiya et al. 1983 [21] found that older chickens have more abdominal fat. When the fat in other parts increased, the intramuscular fat content decreased. An increased SFT and AFW indicated that lipid synthesis in liver was enough. The IMF decrease was not due to insufficient lipid synthesis. Damon et al. 2006 [22] found that the IMF was related to the change of the number of adipocytes. Li et al. 2020 [23] confirmed that genes related to muscle fiber hypertrophy were upregulated while genes related to cell cycle and cell proliferation were inhibited with age in chicken breast muscle. In our study, as intramuscular fat decreases, the diameter of the BMF increases rapidly. Therefore, the reason for the decline in IMF might be that the fat absolute quantity is maintained, but with the rapid growth of muscle fibers, the proportion of fat cells decreases.

### 4.2. Differentially Abundant Proteins in the Liver of Muscovy Duck

To further understand the trait of growth performance at the molecular level, a proteomic analysis of Muscovy duck in liver was conducted. Changes in liver protein abundance can effectively reflect changes in metabolism. In our study, a significant change in liver protein abundance of Muscovy ducks occurred at 28 to 42 days of age. For lean Pekin duck, the change of liver protein abundance occurred at 14 to 28 days of age [13]. Proteins’ abundance change in different periods in the two genera might be because of the differential growth rate.

Rapid growth requires vigorous metabolism. The GO analysis showed that DEPs were mostly enriched in the carboxylic acid metabolic process and single-organism catabolic process at 28 vs. 42 stage in Muscovy duck. In this research, the proteins IDH, DLD, MDH, and PDH were all involved in the carboxylic acid metabolic process, and they were all upregulated at the 28 vs. 42 stage. As rate-limited enzymes during carboxylic acid metabolism, the increased expression suggested an elevated liver metabolism in Muscovy duck. Moreover, the activity of IDH in rat increases with age before adulthood [24]. Different IDH protein spots with the same molecular weight (MW) could be due to the combination of Mg^2+^ and Mn^2+^. DLD and PDH are members of the pyruvate dehydrogenase complex [25]. The tight combination of DLD and PDH might cause the shift of the MW and PI. Different spots all representing MDH could be the isoenzymes of c-MDH and m-MDH in rat [26].

The single-organism catabolic process involved liver detoxification, cholesterol synthesis, folate cycle, and amino acid metabolism. It was found that TST, GS, methyl mercaptan (MM) oxidase, and Aldo-keto reductase family 1 member A1 (AKRA1) participated in liver detoxification for H2S, NH3, MM, and toxic aldehyde [27,28,29]. MM was released by intestinal microbes with an unpleasant odor and could cause halitosis, which is usually caused by intestinal food accumulation [30]. In this study, TST, MM oxidase, and AKR1A1 were upregulated while GS was downregulated at 42 days of age, which might indicate an overfeeding and increased feces accumulation in the intestine [31]. As reported by Yabe-Nishimura. [32], AKR was stimulated by high glucose and opened the polyol pathway when hexokinase was saturated in the glucose metabolism pathway, which further demonstrated that Muscovy duck was overfed at 42 vs. 28 stage. The 5-formyltetrahydrofolate cyclase belongs to the folate cycle and provides a one-carbon unit for the synthesis of proteins, nucleotides, pantothenic acid, and methylation of molecules, thus promoting animal growth and development in rat [33]. The upregulation of FAH might be responsible for the degradation of phenylalanine or tyrosine at 42 days of age. HMG-CoA reductase is the first and crucial step of the catalytic enzyme for the cholesterol synthesis in humans [34], and it is upregulated, suggesting an increased cholesterol synthesis at this stage.

Waterfowl have a natural tendency to overfeed [35], and Muscovy ducks tend to store fat in their livers. Increased sugar and lipid metabolism bring the risk of hyperlipidemia and fatty liver, especially the upregulation of MDH and HMG-CoA reductase, which is the marker enzyme of lipid metabolism in rat [36]. The higher expression of MDH could lead to rapid lipid synthesis and fatty liver formation. At 56 days of age, fatty liver appeared in Muscovy ducks, showing enlargement, yellowing, easy rupture, and increased fragility [37]. The significantly increased expression of ATP5B at 56 days of age might participate in the activation of the PI3K/Akt pathway to alleviate the damage of hyperlipidemia in mice [38].

Generally, the higher the metabolic rate of an organism, the greater the production of reactive oxygen species [39]. In this experiment, the GO terms of differentially abundant proteins were significantly enriched in oxidative stress, especially at early and middle stages, and were all upregulated, which might be responsible for the resistance of oxidative damage and rapid growth. The increase in the final liver weight after force-feeding is always associated with a rise in the cellular oxidative stress level in mule duck [40]. CAT, PRDX1, and PRDX3 directly participate in antioxidant defense [41,42]. Overexpression of mitochondrial-targeted CAT could prevent muscle insulin inhibition caused by a high-fat diet through intramuscular fat degradation in mice [43]. In this study, the decrease of IMF may also be due to the increased expression of CAT. Comparing the IMF content of broilers under different rearing system, lower stocking density and higher exercise of broilers with a floor rearing system were associated with a higher IMF, while broilers reared in cages showed a lower IMF and higher oxidative stress [44], which suggested that the IMF content is negatively associated with oxidative stress. Through PPI analysis, CAT, TPI1, and PDIA3 play a key role in the growth and development of Muscovy ducks. The PDIA3 is a molecular chaperone that catalyzes the disulfide bond formation and isomerization of secreted proteins. The high expression of PDI indicates rapid cell growth and proliferation [45]. The identified key proteins CAT, TPI1, and PDIA3 further suggest the significance of oxidative stress and metabolism during duck growth.

## 5. Conclusions

The present study provided a profile of the fat deposition and liver proteome of Muscovy duck. The stage of 28 to 42 days is a critical period for the growth of Muscovy duck, in which body weight and muscle fibers grow quickly, abdominal fat and subcutaneous fat are deposited rapidly, and intramuscular fat significantly decreases. Liver metabolism and the antioxidant capacity are enhanced with rapid growth. Proteins CAT, TPI1, and PDIA3 are the key candidates responsible for the regulation of fat deposition and growth in Muscovy duck.

## Figures and Tables

**Figure 1 animals-11-02180-f001:**
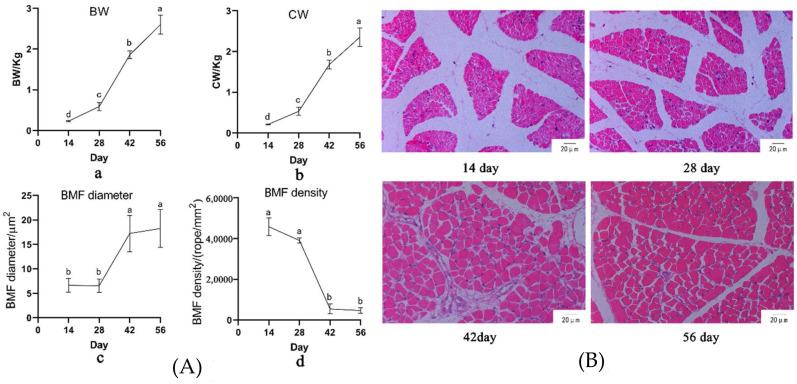
(**A**) Body weight (BW, **a**), carcass weight (CW, **b**), breast muscle fiber (BMF) diameter (**c**), and BMF density (**d**); (**B**) Hematoxylin-eosin (HE) section of breast muscle at different stages.

**Figure 2 animals-11-02180-f002:**
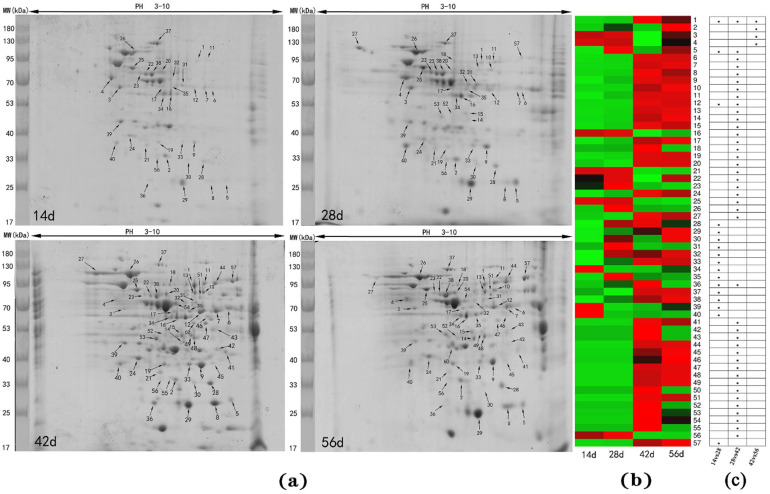
(**a**) Two-DE profiles of liver in Muscovy duck, day (d); (**b**) The relative expression of differentially abundant proteins at different stages, the gradient from green to red represents a gradual increase in protein expression; (**c**) The *p*-values of spots at 14 vs. 28, 28 vs. 42 and 42 vs. 56, *p* < 0.05 = *.

**Figure 3 animals-11-02180-f003:**
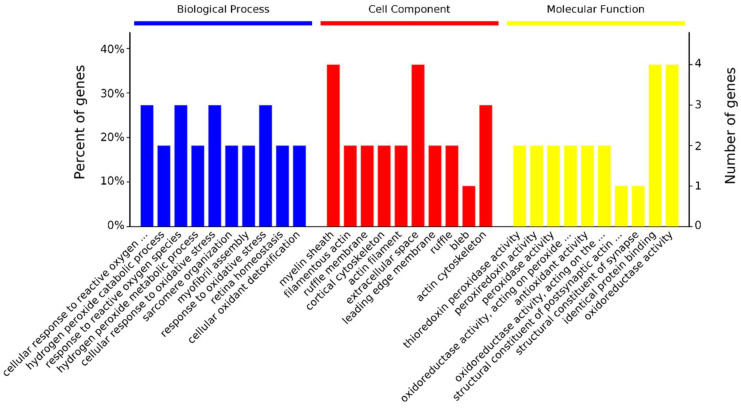
GO annotation of 13 differentially abundant proteins in three categories: biological process, cellular component, and molecular function at 14 vs. 28 day. Only the terms with *p*-values less than 0.05 are listed.

**Figure 4 animals-11-02180-f004:**
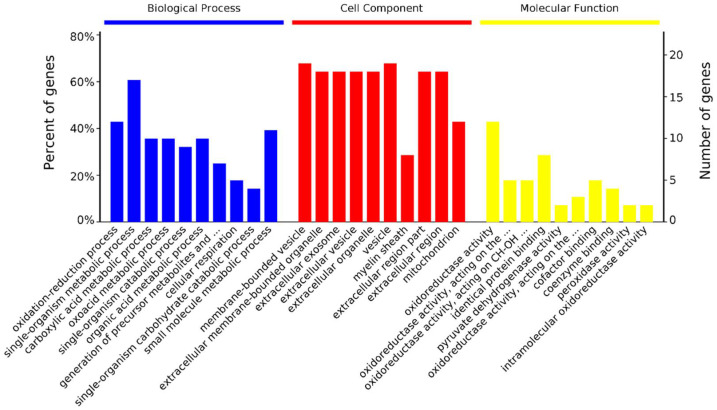
GO annotation of 39 differentially abundant proteins in three categories: biological process, cellular component, and molecular function at 28 vs. 42 day. Only the terms with *p*-values less than 0.05 are listed.

**Figure 5 animals-11-02180-f005:**
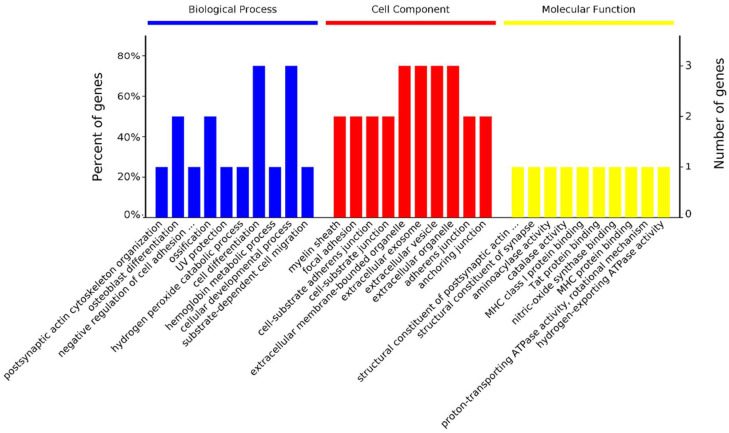
GO annotation of 4 differentially abundant proteins in three categories: biological process, cellular component, and molecular function at 42 vs. 56 day. Only the terms with *p*-values less than 0.05 are listed.

**Figure 6 animals-11-02180-f006:**
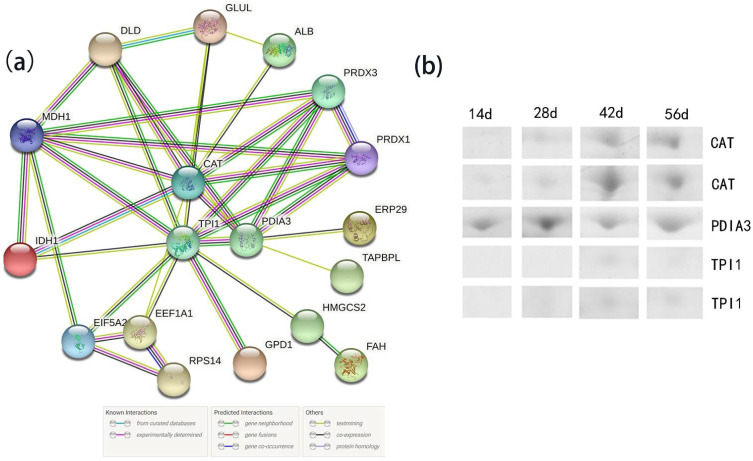
(**a**) The biological interaction network of the proteins for all differentially expressed proteins; (**b**) The expression level of node proteins in 2D electrophoresis gels, day (d). Note: CAT, catalase; TPI1, triosephosphate isomerase; PDIA3, protein disulfide-isomerase; ERP29, endoplasmic reticulum protein 29; TAPBPL, glyceraldehyde-3-phosphate dehydrogenase; HMGCS2, 3-hydroxy-3-methylglutaryl coenzyme A synthase; FAH, fumarylacetoacetase; GPD1, glycerol-3-phosphate dehydrogenase; EEF1A1, elongation factor 1-alpha; RPS14, ribosomal protein S14; EIF5A2, eukaryotic translation initiation factor 5A; IDH1, isocitrate dehydrogenase; MDH1, malate dehydrogenase; DLD, dihydrolipoyl dehydrogenase; GLUL, glutamine synthetase; ALB, albumin; PRDX3, Peroxiredoxin 3; PRDX1, Peroxiredoxin 1.

**Table 1 animals-11-02180-t001:** Fat deposition of Muscovy duck at different periods.

Age (Day)	IMF ^1^ %	SFT ^2^ (mm)	AFW ^3^ (g)
14	/	1.37 ^b^ ± 0.252	0.45 ^c^ ± 0.370
28	2.89 ^a^ ± 0.65	1.62 ^b^ ± 0.274	3.31 ^c^ ± 1.73
42	1.15 ^b^ ± 0.27	2.50 ^a^ ± 0.556	28.23 ^b^ ± 6.95
56	1.25 ^b^ ± 0.36	2.38 ^a^ ± 0.557	44.23 ^a^ ± 12.189

^a,b,c^ Date of the same column with different superscripts indicate significant differences (*p* < 0.05). ^1^ IMF, the percentage of fat in muscle. ^2^ SFT, the thickness of subcutaneous fat near the pectoral muscle. ^3^ AFW, the abdominal fat weight.

**Table 2 animals-11-02180-t002:** List of significant differentially abundant proteins in Muscovy ducks between the three stages.

Sort	Spot ^1^	Accession ^2^	Proteins Description	S ^3^	M ^4^	SC ^5^ (%)	MW (kDa)/pI ^6^	
							Thero.	Obs.
Enzymes and catalytic function	1	A0A493TQM7	Catalase (CAT)	127	12	14.8	60.2/7.7	95.0/7.9
	11	A0A493TQM7	Catalase (CAT)	92	8	9.3	60.2/7.7	95.0/8.2
	6	U3J597	Isocitrate dehydrogenase [NADP] (IDH)	439	24	27.5	47.0/7.6	70.0/9.0
	7	U3J597	Isocitrate dehydrogenase [NADP] (IDH)	467	24	27.5	47.0/7.6	70.0/8.7
	9	U3I1N5	Sulfurtransferase (TST)	187	15	25.2	33.1/7.1	39.0/8.2
	42	U3I1N5	Sulfurtransferase (TST)	70	3	8.8	33.1/7.1	41.0/8.7
	10	U3IR48	Dihydrolipoyl dehydrogenase (DLD)	932	35	22.5	50.1/8.4	82.0/8.2
	13	U3IR48	Dihydrolipoyl dehydrogenase (DLD)	220	8	12.8	50.1/8.4	82.0/8.0
	51	U3IR48	Dihydrolipoyl dehydrogenase (DLD)	88	3	7.4	50.1/8.4	72.0/8.1
	12	A0A493SW39	Fumarylacetoacetase (FAH)	221	13	16.2	46.7/7.2	68.0/8.2
	14	A0A493T656	Glycerol-3-phosphate dehydrogenase [NAD(+)] (GPD1)	57	7	10.3	39.5/8.2	43.0/7.9
	56	A0A493TUY2	Glyceraldehyde-3-phosphate dehydrogenase (TAPBL)	63	7	10.6	42.2/5.3	34.0/7.4
	15	U3IR57	Malate dehydrogenase (MDH1)	115	9	19.8	36.7/6.9	45.0/7.8
	53	U3IR57	Malate dehydrogenase (MDH1)	245	11	17.7	36.7/6.9	44.0/7.1
	16	A0A493T5Z6	Glutamine synthetase (GS)	211	12	11.4	52.0/7.6	55.0/7.7
	17	A0A493T5Z6	Glutamine synthetase (GS)	123	4	3.5	52.0/7.6	55.0/7.6
	48	A0A493T5Z6	Glutamine synthetase (GS)	55	2	4.5	52.0/7.6	49.0/8.0
	19	A0A493TH73	Quinoid dihydropteridine reductase (QDPR)	187	2	15.5	28.2/6.1	39.0/7.3
	20	A0A493TX20	Methyl mercaptan (MM) oxidase	57	6	9.3	57.2/7.2	75.0/7.2
	25	U3IS54	Protein disulfide-isomerase (PDIA3)	615	47	38.1	56.6/5.8	80.0/6.3
	33	A0A493U078	Peroxiredoxin 3 (PRDX3)	146	8	12.9	29.1/9.3	39.0/7.8
	36	U3II01	Peroxiredoxin 1 (PRDX1)	193	11	19.1	22.6/8.2	27.0/7.0
	34	A0A493STH1	Pyruvate dehydrogenase E1 component subunit alpha (PDH)	226	11	16.9	45.4/8.2	58.0/7.5
	46	A0A493STH1	Pyruvate dehydrogenase E1 component subunit alpha (PDH)	332	15	20.8	45.4/8.2	58.0/8.7
	45	U3I8D8	Triosephosphate isomerase (TPI1)	233	11	27	24.2/7.7	40.0/8.5
	52	U3I8D8	Triosephosphate isomerase (TPI1)	387	16	26.1	24.2/7.7	49.0/7.3
	47	U3II47	Aldo-keto reductase family 1 member A1 (AKR1A1)	285	16	16.5	37.3/7.2	54.0/8.5
	50	U3II47	Aldo-keto reductase family 1 member A1 (AKR1A1)	74	4	7.6	37.3/7.2	54.0/8.1
	49	U3J9Y2	L-lactate dehydrogenase (LD)	360	16	24.3	36.5/7.1	52.0/8.0
	54	A0A493T8G2	3-hydroxy-3-methylglutaryl coenzyme A synthase (HMGCS2)	231	9	7.5	60.6/8.7	72.0/7.5
	55	U3IT75	5-formyltetrahydrofolate cyclo-ligase	74	9	21.6	24.1/6.4	34.0/7.0
Structural and functional proteins	2	A0A493TSC9	Zinc finger and BTB domain containing22 (ZBTB22)	70	2	1.7	57.5/6.6	34.0/7.5
	27	A0A493TSC9	Zinc finger and BTB domain containing 22 (ZBTB22)	43	3	1.7	23.2/5.2	98.0/5.0
	4	U3IP65	ATP synthase subunit beta (ATP5B)	1087	54	38.7	53.5/5.2	75.0/5.7
	18	U3J2L0	Ribosomal protein S14 (RPS14)	66	1	7.1	16.6/8.9	93.0/7.4
	22	U3IL97	Keratin 12 (KRT12)	123	6	8.4	49.9/5.0	75.0/6.7
	23	U3IM27	Aldedh domain-containing protein	141	12	17.1	46.7/5.7	75.0/6.5
	38	U3IM27	Aldedh domain-containing protein	314	22	19.9	46.7/5.7	75.0/6.9
	24	A0A493TAD1	Eukaryotic translation initiation factor 5A (EIF5A2)	130	5	5.6	15.9/5.5	42.0/6.4
	57	A0A493SUB8	Elongation factor 1-alpha (EEF1A1)	100	3	4.1	50.5/9.1	96.0/9.1
	26	A0A493TIR6	Albumin (ALB)	2282	144	62.5	69.3/5.9	98.0/6.2
	29	A0A493T6B1	IF rod domain-containing protein	93	3	3.4	52.7/4.8	26.0/7.9
	43	A0A493T6B1	IF rod domain-containing protein	102	4	3.4	52.7/4.8	53.0/8.7
	30	A0A493TP63	IF rod domain-containing protein	71	4	5.8	55.3/4.9	33.0/8.1
	32	A0A493U1E6	Histone H2B	93	3	15.1	14.0/10.3	70.0/7.8
	37	A0A493TBB4	Ovotransferrin	328	25	18.8	82.6/6.9	130.0/7.0
	39	A0A493SXF3	Tropomyosin 4 (TPM4)	33	1	2.4	28.8/4.6	45.0/6.0
	41	U3ILS4	Endoplasmic reticulum protein 29 (ERP29)	207	19	26.8	29.2/8.6	41.0/9.2
	44	U3J7T0	Myosin motor domain-containing protein	1588	102	30.3	187.8/5.4	95.0/8.8
Uncharacterized proteins	5	A0A493SU68	Uncharacterized protein	66	19	10.5	23.3/5.2	27.0/9.0
	21	A0A493SU68	Uncharacterized protein	70	3	10.5	23.2/5.2	34.0/7.2
	31	A0A493SU68	Uncharacterized protein	70	2	10.5	23.2/5.2	70.0/8.1
	35	A0A493SU68	Uncharacterized protein	76	2	10.5	23.2/5.2	69.0/8.2
	8	A0A493TMK3	Uncharacterized protein	756	43	29.4	37.1/9.3	26.0/8.5
	3	A0A493T8E7	Uncharacterized protein (ACTG1)	641	44	34.6	42.2/5.3	70.0/6.1
	40	A0A493T8E7	Uncharacterized protein (ACTG1)	87	5	4.5	42.2/5.3	40.0/6.1
	28	A0A493TX61	Uncharacterized protein	261	26	30.8	20.9/5.8	39.0/8.4

^1^ Spot ID represents the protein spot number on the 2-DE gel image. ^2^ Accession numbers of matched proteins according to the UniProt protein database refers to the species *Anas platyrhynchos*. ^3^ Score, the MASCOT score for which ion scores as a nonprobabilistic basis for ranking protein hits. ^4^ Matched, matched peptides for the number of pairings in an experimental fragmentation spectrum to a theoretical segment of protein. ^5^ Sequence coverage (%), the ratio of the number of amino acids in every peptide that matches with the mass spectrum in the total number of amino acids in the protein sequence. ^6^ Theoretical molecular weight (MW) and isoelectric point (pI) of the identified proteins retrieved from the UniProt protein database.

**Table 3 animals-11-02180-t003:** The number of significant spots and proteins.

	14 vs. 28 Day	28 vs. 42 Day	42 vs. 56 Day
Significant spots	17	41	4
Significant proteins	13	39	4

## Data Availability

The mass spectrometry proteomics data have been deposited to the ProteomeXchange Consortium (http://proteomecentral.proteomexchange.org) via the iProX partner repository with the dataset identifier PXD025920.

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
