# Peer review of "Hepatic Proteomic Analysis Reveals That Enhanced Carboxylic Acid Metabolism and Oxidoreduction Promote Muscle and Fat Deposition in Muscovy Duck"

_animals, 2021, doi:10.3390/ani11082180_

Round 1

Reviewer 1 Report

The paper entitled « Proteomic analysis reveal enhanced carboxylic acid metabolism and oxidoreduction promoted muscle and fat deposition in Muscovy duck » aims to study the evolution of Muscovy ducks during their growth. Ducks were studied and sampled at 14, 28, 42 and 56 days of age. In a first part the body weight and the main energy storage tissues are studied and in a second part the liver proteins are studied to better understand the development of Muscovy ducks that are slow growth rate meat-type ducks.
The animal experimental design, the protein analyses, the statistical analyses and the biological interpretation are relevant.
Some minor revisions are required. There are carefully described in the enclosed document.

Reviewer 2 Report

The present study provided a profile of the fat deposition and liver proteome of Muscovy duck. The number of birds (n=8 for date evaluation) used in the experiment is sufficient. The test methods used are correct, but Materials and Methods chapter requires additions. The discussion is well carried out and exhausting. References well chosen. Before publishing in Animals, the paper requires additions and corrections. The list of proposed changes is given below:

General comments:

Please prepare the article according to the instructions for the authors.

For affiliates, the first name and surname initials for each co-author of the article should be provided, the same as given in the "Author Contributions" chapter. For example, W.Y. instead of YW; C.W. instead of WC in the Author Contributions chapter

For significance description please use lowercase p in italic, spaces before and after “<” for example (p < 0.05) – L29, 174, 176, 185 etc.

In the References chapter, abbreviated name journal must be used, the volumen number must be in italic

For a page range use a long "-" from the insert function for all References items

Detailed comments:

L5 "1" superscript for „Liu”

L11 No initials "X.C" for Xingyong Chen

L16 "Muscovy" with a capital letter

L16 and other "28 vs 42" days of age, space after 28 and before 42

L17 „subcutaneous” instead of subcutaneou, add "s"

L22, 35 "ducks" instead of "duck"

L31 GO (full name) instead of current form

L49 add Reference: Chartrin et al. (2006) from Poultry Science on IMF

L50 add Reference: Kokoszyński et al. (2021) in Journal Applied Animal Resarch containing information on the date of slaughter of Muscovy ducks and Pekin broilers

L55 and other "Pekin" instead of "Peking"

L79 No purpose of the experiment (article)

L89 add information about: total number of Muscovy ducks, sex of birds (males?), type of building (closed, without windows) and type of floor (deep litter?)

L90 + add information about the microclimate parameters (temperature from to; humidity from to), photoclimate (length, color, intensity)

L90 add information about the content of CP and MJ ME, as well as the energy to protein ratio, the content of choline - that determines the fatness of the carcass.

L99 enter the name of the balance, manufacturer's data, measurement accuracy

L100 provide the name of the instrument and manufacturer data for the determination of IMF content

L103 add information on the method of sampling and preparation of histological slides

L180 I propose: “Figure 1. Body weight (BW), carcass weight (CW), breast muscle fiber (BMF) diameter, and breast muscle fiber (BMF) density (a), and……….. (HE) section of… .. instead of currrent form

L188 I propose "Table 1. The fatness of Muscovy duck ... .."

L234, 245 GO - what is it?

L280 "The Catalase (CAT)" instead of The CAT

L300 „breeds” instead of species, broilers is not a species but a production direction

L300-301 add Murawska et al. (2012) about growth rate of tissue and organs in Pekin ducks in Poultry Science instead of „Motoko et al.”

L302 Sonaiya et al. [17], removing [17] after "fat"

L305 Damon et al. [18]

L306 remove „[18]” after adipocytes

L389 C.X. instead of CX, etc.

Refrences = see Genetal comments

L408 ".... TopoliÅ„ski, T; Andryszczyk, M; Wirwicki, M. " - lost „Andryszczyk, M.”

Reviewer 3 Report

Peer review:

Manuscript Title:  Proteomic analysis reveal enhanced carboxylic acid metabolism and oxidoreduction promoted muscle and fat deposition in Muscovy duck

Journal: Animals

Number: animals-1296167

To the editor and the authors:

It is an interesting article about fat metabolism in ducks. The research question is important to improve the knowledge of Muscovy duck metabolism in different farming ages. It is also necessary to provide scientific information about proteomic changes of liver tissue in ducks. The manuscript provides experimental molecular evidences of protein occurrence and relationship with some biochemical consequences in ducks submitted to a regular diet with nutrition level under the National Research Council (NRC) standard. Such scientific article is important to all scientists and professionals working in the poultry production chain.

The whole article is well presented with appropriate Tables and Figures. The introduction is complete, the experimental animals and the methods are well detailed, the results present the main findings and the Discussion is clear. I have just missed a more detailed Discussion about the importance of the study for duck farming. It would be important to highlight the contribution of the study to provide the basis for developing a reasonable diet for Muscovy duck (as written in the Simple Summary). One or two paragraphs about this subject would be very welcome. There are also some minor errors that need to be solved in the whole text. And a careful English revision is necessary.  

Minor modifications:

1)        Simple Summary: I would recommend to remove the last sentence (“and further provides the basis for developing a reasonable diet for Muscovy duck”) if the authors do not include a paragraph in the Discussion about this subject.

2)        Methodology (lines 90-92 and 121): “Eight healthy ducks at the age of 14, 28, 42, and 56 days, respectively, were randomly selected for sampling as eight four replicates.” “Eight samples from each time-point were used as 8 four replicates for…”.

3)        Results (lines 234-254): it is not necessary to divide in the subtopics 3.4.1 and 3.4.2. Please include the two paragraphs in only one topic (3.4).

Reviewer 4 Report

Line 53  after  surrounding tissues. add “Also in geese overfed the amount of synthetic fat is much more than the amount of fat transported, causing a large accumulation of fat in the liver to form fatty liver (3)”

(3) Xu Liu, X.;  Liu, Y.; Cheng, H.; Deng, Y.; Xiong, X.; Qu, X. Comparison of performance, fatty acid composition, enzymes and gene expression between overfed Xupu geese with large and small liver. It. J. Anim. Sci. 2021, 20, 102–111.

At the end of the introduction, the sentence must indicate the aim of the work.

Line 88 change “The Muscovy ducks” as “The male Muscovy ducks”

Lines 90-92 “Eight healthy ducks at the age of 14, 28, 42, and 56 days, respectively, were randomly selected for sampling as eight replicates.”  Unclear sentence. Were The Muscovy ducks 32? or were there replicates? How many Muscovy ducks have been weighed to calculate Body weight (BW), carcass weight (CW) ?

Line 103 “The total number of fibers was calculated (N)”  This parameter is not reported in the results.

Line 121 delete “as 8 replicates”

Line 178 change  “(Figure 1b)” as  “ (Figure 1)”

Lines 300-301   delete “Motoko et al. indicated that 300 subcutaneous fat increased with age-dependent before 20 years old in human beings [16].”
